# The Development of an Oral Health Nursing Tool for Patients with a Psychotic Disorder: A Human-Centred Design with a Feasibility Test

**DOI:** 10.3390/ijerph21040492

**Published:** 2024-04-17

**Authors:** Sonja Kuipers, Stynke Castelein, Linda Kronenberg, Job van ’t Veer, Nynke Boonstra

**Affiliations:** 1Research Group Healthcare & Innovation in Psychiatry, Department of Healthcare, NHL Stenden University of Applied Sciences, Rengerslaan 8-10, 8900 CG Leeuwarden, The Netherlands; nynke.boonstra@nhlstenden.com; 2Department of Experimental Psychopathology and Clinical Psychology, Faculty of Behavioural and Social Sciences, University of Groningen, Grote Kruisstraat 2/1, 9712 TS Groningen, The Netherlands; s.castelein@lentis.nl; 3Lentis Research, Lentis Psychiatric Institute, Hereweg 80, 9725 AG Groningen, The Netherlands; 4Dimence Mental Health Care, Burgemeester Roelenweg 9, 8021 EV Zwolle, The Netherlands; l.kronenberg@dimencegroep.nl; 5Research Group Digital Innovation in Care and Welfare, Department of Healthcare, NHL Stenden University of Applied Sciences, Rengerslaan 8-10, 8900 CG Leeuwarden, The Netherlands; job.van.t.veer@nhlstenden.com; 6KieN Early Intervention Service, Oosterkade 72, 8911 KJ Leeuwarden, The Netherlands; 7UMC Utrecht Brain Center, University Medical Center Utrecht, Heidelberglaan 100, 3584 CX Utrecht, The Netherlands

**Keywords:** oral health, nursing, mental health, psychotic disorder, human-centred design

## Abstract

Patients with psychotic disorders frequently report oral health problems, while mental health nurses (MHNs) seem not to be fully aware of these problems and the risk factors. Therefore, this study aimed to develop supportive tools for MHNs regarding oral health to increase sensitisation among MHNs and provide MHNs with the knowledge to recognise (potential) oral health problems in patients with a psychotic disorder. We used a human-centred design in which the user, MHNs and experts by experience were placed at the centre of the research process. Problems and needs in MHNs working with patients with a psychotic disorder were addressed. To identify key issues of problems as well as needs in terms of resources, we started with focus groups (*n* = 9). We analysed the data thematically based on the context of patients and MHNs in practice regarding oral health, preferred oral health tools focused on MHNs, and the intended outcomes of tools. A multi-criteria decision matrix was developed and analysed (*n* = 9) to identify the most optimal and viable solution based on established criteria and issues that are prevalent in the work of MHNs. The most promising result was the development of a brochure with an awareness screener. The brochure with the awareness screener was developed as a low-fidelity prototype for MHNs regarding oral health in patients with a psychotic disorder based on the latest scientific evidence. After testing it, the feasibility was tested through semi-structured interviews (*n* = 19). MHNs and experts by experience were satisfied with the tool and provided recommendations for adjustments to the content. Significant augmentations to the brochure included a clinical lesson and a toothbrush with toothpaste for patients. We can conclude that a brochure with an awareness screener is feasible for MHNs. Future steps aiming to further refine and optimise care processes for oral health tools in MHNs should take refining eligibility criteria for psychiatric populations and the language level of the target group of a tool into account.

## 1. Introduction

Since the World Health Organization emphasised that oral health is integral and essential to general health and well-being [1,2], oral health has improved in the general population; however, vulnerable patients are an exception [3]. Epidemiological studies have shown that the lifespan of patients with psychotic disorders is shorter compared to that of the general population without mental illness. An unhealthy lifestyle is an important cause of the 15-to-25-year shortened life expectancy for patients diagnosed with (severe) mental illness. Several studies of patients with (severe) mental illness have shown that oral health and oral health-related quality of life are substandard among those with an unhealthy lifestyle [4,5,6,7,8,9].

Inadequate oral health self-management, lower tooth-brushing frequency, and a lack of motivation to perform proper oral hygiene are known barriers to adequate oral health for patients with psychotic disorders [10,11,12,13]. Roebroek et al. [14] demonstrated that psychiatric care needs were well discussed by psychiatrists, psychologists, and mental health nurses (MHNs), but treatment plans paid little attention to physical care needs and needs related to social well-being. This was supported by Happell et al. [15], who reported that when patients express concerns regarding their physical health, health care professionals tend to pay little attention to them [15]. The literature shows that a healthy lifestyle and behavioural changes are difficult to initiate and even more challenging to sustain over time [16]. This highlights the disconnection that can occur in mental health care settings between patients’ concerns and responses from MHNs, as well as the inherent difficulties in altering one’s lifestyle to become healthier, both in the initiation phase and in maintaining these lifestyle changes permanently. One explanation is that psychiatric treatment and care have historically been provided mainly based on the medical model, which focuses on alleviating the symptoms of illness (e.g., by giving medication), despite the fact that nursing should be provided from a holistic point of view. In the 19th century, when nursing was defined for the first time, Florence Nightingale articulated the concept of the science of holistic nursing [17,18]. This concept of a holistic vision of care underpins the profession of MHNs; it is important that the integration of physical issues, such as oral health, is further addressed. Unfortunately, we see this too little in clinical practice. 

Since 2011, health has been defined as “the ability to adapt and self-manage, in light of the physical, emotional and social challenges of life” [19]. In this definition, which is widely accepted, health is no longer seen as the absence or presence of disease but as the ability to deal with (changing) physical, emotional, and social life challenges and manage oneself as much as possible. This implies a vision that fits with holistic nursing. From a holistic perspective [20], supporting the general health of patients with psychotic disorders, including their oral health, is one of the tasks of MHNs. MHNs are at the forefront of everyday care services [9,15]. However, MNHs indicate that they would like to be more aware of the oral health of their patients [9,21], but they hesitate to take action in this regard due to a lack of both relevant expertise and the practical tools necessary for supporting patient oral health [9]. Relevant expertise is lacking, and there is also a lack of practical tools for MHNs to use when supporting patients with their oral health. Oral health issues are important for everyone; however, patients with psychotic disorders are known to have related risk factors (e.g., smoking) that contribute to a lower oral health-related quality of life and are possibly associated with a shorter life expectancy [4,5,6,7,8].

Recent studies [7,21] show that a wide range of interventions must be developed for MHNs to support patients with psychotic disorders in maintaining and improving their oral health. Initially, MHNs must increase their sensitivity and alter their behaviour concerning oral health. To our knowledge, no studies have yet been conducted on the sensitisation of MNHs regarding maintaining and improving the oral health of patients with psychotic disorders. There are many models in the field of behavioural change, and Prochaska and DiClemente’s transtheoretical model of change is particularly suitable here because it shows the different phases of behavioural change [22]. The transtheoretical model’s broad applicability and focus on individual readiness make it a valuable model for understanding and facilitating change in MHNs, with a supportive tool to increase their sensitisation and provide them with the knowledge to recognise potential oral health problems in patients with psychotic disorders. Prochaska and DiClemente developed this model as a way to integrate the stages and processes of behavioural change by dividing it into six stages. Our previous study showed that MHNs act mainly in the first two stages of the model [21,23]: (1) pre-contemplation stage: MHNs do not intend to make behavioural changes in the foreseeable future; (2) contemplation stage: MHNs consider a behavioural change but do not yet make a firm commitment to change. To promote behavioural changes among MHNs, it is important to first increase their sensitisation regarding oral health care in their patients. Developing a supportive tool based on the Prochaska and DiClemente model involves understanding the nuances.

The aim of this study was to develop a supportive tool for MHNs regarding oral health to increase their sensitisation and provide them with the knowledge to recognise potential oral health problems in patients with psychotic disorders. 

## 2. Methods

### 2.1. Study Design

For this study, a qualitative human-centred design (HCD) approach was used to develop a tool that serves the dual purpose of raising sensitisation among MHNs and providing them with the knowledge to recognise oral health problems in patients with psychotic disorders. One widely recognised HCD framework for innovative design projects is the double diamond (DD) framework (Figure 1), originally devised by the Design Council [24]. This double diamond framework directs problem-solving thought processes within the HCD methodology. In the context of the present study, the focus was primarily on the “develop” and “deliver” phases of the DD framework, with the overarching objective of creating a tool that would effectively address key problem area [25], i.e., increase sensitisation and knowledge among end users, in this case, MHNs. The HCD approach utilised in this initiative, characterised by its participatory and iterative nature, departs from conventional methodologies by involving close collaboration between MHNs and experts by experience to gain a better understanding of patients’ specific needs, which enables the design of tailored tools and provides MHNs with a significant role in guiding the development of solutions. Tools targeting MHNs were also indirectly linked to tools targeting patients with psychotic disorders. Therefore, the involvement of experts by experience was imperative to represent the patient perspective. Within the framework of this study, we employed three iterations (the first focused on developing the multi-criteria decision matrix (MCDM), the second aimed at creating an oral health tool for MHNs, and the third aimed at evaluating the feasibility of the oral health tool for MHNs). 

The iterative stages and data collection activities within the DD research process are depicted in Figure 1. 

### 2.2. Study Population

For step 1a in our research process (pressure cooker focus group session), we recruited nine participants (four MHNs and five experts). To achieve maximum variation in our sample, participants were selected based on purposive sampling by the research team based on their knowledge and experience with patients with psychotic disorders and/or oral health problems. Participants were approached by e-mail, telephone, or face-to-face. For this focus group session, we included four professionals: one master’s-level advanced nurse practitioner and three bachelor’s-level MHNs, based on their knowledge of and experience with patients with psychotic disorders. Furthermore, we included five other specialists: an oral health hygienist who also had a master’s in health innovation was included based on her knowledge of and experience with oral health and human-centred design; another expert was included based on their experience with psychotic disorders and oral health problems; an education specialist was included based on her expertise in and experience with how to best transfer knowledge; and, finally, it was important to include a communication specialist based on their knowledge of and experience with conveying messages for the purpose of encouraging behavioural changes in MHNs.

In step 2a, the participants included in step 1a were invited to score the 15 prototype tools of the MCDM.

In step 3a, we included professionals: master’s-level advanced nurse practitioners, bachelor’s-level MHNs, and experts by experience based on convenience sampling. Participants were included based on their experience (in working years) with in- or outpatients with psychotic disorders, availability, and willingness to participate. The iterative process of sampling, data collection, and analysis was continued until data saturation was reached; no new codes were found in the last four interviews. Participants were approached by e-mail and recruited from the KieN Early Intervention Service (Leeuwarden), Lentis Psychiatric Institute (Groningen), Dimence psychiatric services (Zwolle), and Friesland Mental Health Care Services (Leeuwarden) in the Netherlands. 

### 2.3. Data Collection

Step 1a: Pressure cooker focus group session

Before the start of the pressure cooker focus group session, participants were split between two groups (focus groups 1 (*n* = 5) and 2 (*n* = 4)), in which the research team ensured an even distribution of knowledge and experience. In terms of content, both groups discussed the same topics at the same time.

First, in step 1a, the session was divided into two parts. The first part was aimed at brainstorming around the question, “how can we make MHNs more sensitive of the urgency of paying attention to the oral care of patients with psychotic disorders?”

Because rapidity adds value to the creative process, we used the format of a pressure cooker session [25,27]. A pressure cooker session is a brief intensive session targeting essential topics in limited time. The benefits of a pressure cooker session are that it can provide (1) a concrete plan and actions, (2) substantial time savings, (3) a concrete plan with actions, and (4) greater support for and commitment to the choices and actions [25,27]. To stimulate creativity, the pressure cooker session was divided into two subgroups. The input for the first part of the session was eight inspiration cards based on metaphors created by the research group (Appendix A). Casakin [28] noted that metaphors can help to identify and capture design concepts, as well as define goals and requirements and stimulate creative thinking. The inspiration cards were based on public campaigns in the Netherlands to raise sensitisation among people. 

The second part of this brainstorming session aimed to identify solutions based on personas, as developed in our previous article [21]. These personas (Anna, Monica, Julia, and Paul) showed a diversity in attitudes and perspectives on oral health from MHNs. Furthermore, in the personas, there were differences in barriers, needs, and suggestions for interventions from MHNs. The personas were prepared by the research team and sent to participants prior to the session, and we asked them to empathise with each persona. The main question was, “considering the characteristics and circumstances of this person, which main problem should be solved and what needs do they have? And, if the MHNs were to meet the person in practice, what should they do and why”? Every session was audio-recorded and transcribed verbatim. 

Step 2a entailed prioritising insights and concepts for prototypical tools aimed at enhancing the decision-making process utilising a multi-criteria decision matrix (MCDM) [29]. The MCDM involved a procedure for identifying the optimal and most viable solution based on established criteria and prevalent issues in the work of MHNs in the context of oral health care for patients with psychotic disorders. The input (15 possible tools) for the MCDM was derived from the analysis of the focus group results in step 1a. Using an MCDM [29] can enhance participants’ decision-making ability by facilitating knowledge transfer, ensuring that all factors relevant to a decision are comprehensively considered and that the rationale behind decisions is clearly communicated. During the initial phase, the 15 possible tools were discussed, and seven criteria were formulated. These criteria, based on the work of Prochaska and Di Clemente (criteria 1–3) [22] and key critical success factors in implementation (criteria 4–7) [30], were that the tool should (1) enhance knowledge pertaining to oral health care among MHNs (not oral health care specialists); (2) contribute to raising the sensitisation of MHNs regarding the importance of oral health; (3) motivate MHNs to engage in specific behaviours (call to action); (4) encourage the adoption of sustainable behavioural changes; (5) integrate seamlessly into existing operational work; (6) be positive; and (7) be implemented rapidly and with ease within organisations. The MCDM was sent to participants by e-mail. Subsequently, all participants (*n* = 9) were asked to assess each tool based on the design criteria using a numerical scale ranging from 1 (low criterion score) to 5 (excellent criterion score). Additionally, participants provided reasons for their scores for each criterion.

In step 3a, the tool was developed as a low-fidelity prototype based on the latest scientific insights from the literature (by SK, NB, and SB). This entailed creating a basic prototype to concretise ideas and facilitate testing [26]. Low-fidelity prototyping is a swift and efficient method by which to convert design ideas into tangible and testable artefacts. The primary and most critical function of low-fidelity prototypes is that they can evaluate and examine functionality, rather than focusing on the visual aesthetics of the product [26]. Participants were asked to critically review and identify areas for improvement in the low-fidelity prototype over 2 weeks. We conducted semi-structured interviews to explore the feasibility of the tested low-fidelity prototype in the field of mental health in a sample of participants who had not yet been involved in this study. Our objective was to evaluate the feasibility of the first low-fidelity prototype and investigate whether the individuals who were expected to be impacted by it were receptive to the tool and whether it would be pragmatic to contemplate its deployment in specific contexts. This could prevent the implementation of an intervention that does not apply to the target population or is, for example, too costly to implement [31]. For this purpose, a six-item topic list was developed based on a study by Bowen et al. on designing a feasibility study [30]. Six factors were identified that could be considered different aspects of feasibility: acceptability, demand, implementation, practicality, integration, and efficacy. Acceptability assesses people’s reactions to a tool, like its suitability and their satisfaction; demand gauges estimated use and interest; implementation examines the scope, likelihood, and methods of comprehensive execution; practicality evaluates feasibility under resource, time, and commitment constraints; integration looks at the extent of systemic changes needed for incorporation; and efficacy focuses on the likelihood of success and the intended effects of the tool [30].

The semi-structured interviews were conducted online utilising Microsoft Teams, business version (Microsoft.com (accessed on 15 December 2023)). Every session was audio-recorded and transcribed verbatim. The data were collected by a trained research nurse (SK).

The steps of data collection and analysis are visualised in Figure 2.

### 2.4. Analysis

In step 1b, a thematic analysis [32] was employed as a guiding framework for the analysis of the transcripts. Data analysis was carried out utilising deductive and inductive thematic approaches to affirm or refute the conclusions [32]. During the development phase, structures and patterns emerged through the process of fragmentation, with open codes being assigned to these fragments. Data coding facilitated the identification of patterns within the data and group statements with thematic similarities [33]. In the reduction phase, the focus was on identifying coherent themes based on the assigned codes, followed by a process of revising and refining these themes. In the final phase, we engaged in critical reflections on the identified themes and overall analysis process [32]. The key themes of the context of patients and MHNs in practice regarding oral health, preferred oral health tools focused on MHNs of patients with a psychotic disorder, and the intended outcomes of tools were discussed as they related to our research questions. Quotes were added to the themes to provide a rich description. The outcomes that emerged from the focus group were used for the MCDM. All data were analysed using Atlas TI version 9 (Atlasti.com (accessed on 15 December 2023)).

In step 2b, a quantitative descriptive analysis of the numerical data derived from the MCDM was performed. Unweighted means were calculated for each participant and each design criterion. Ultimately, these were calculated for each tool. Finally, the full range of options and their cumulative scores were thoroughly discussed (by SK, JV, and NB), culminating in decisions made regarding the tool to be developed.

In step 3b, the results of the semi-structured interviews were deductively analysed using the steps of the thematic analysis (see step 1b) [32]. The results were presented based on the different aspects of feasibility. The key themes of acceptability, demand, implementation, practicality, integration, and efficacy were defined a priori for the analysis in step 3a.

### 2.5. Ethical Considerations

Standard rules governing good clinical practice and ethical principles, originating from the Declaration of Helsinki, were adhered to by informing all participants about the study and their rights and obtaining written informed consent from all subjects [28].

Permission for this research was obtained from the scientific research committees within the organisations KieN Early Intervention Service (Leeuwarden), Lentis Psychiatric Institute (Groningen), Dimence psychiatric services (Zwolle), and Friesland Mental Health Care Services (Leeuwarden), the Netherlands. In compliance with international safety regulations for data retention, all data from the design sessions will be stored for a period of 10 years at NHL Stenden University of Applied Sciences and will only be accessible to three of the authors (SK, NB, and JV).

## 3. Results

### 3.1. Participants and Outcome Focus Groups: Step 1

Step 1a took place in September 2023. For this first design session, we included four professionals and five specialists (*n* = 9). The total group consisted of seven women and two men; their ages ranged between 25 and 53 years (mean age: 40.5 years). Participant characteristics (profession, gender, age, educational level, and years of working experience) are given in Table 1. The pressure cooker session for focus groups 1 and 2 had a duration of 100 min (or 200 min in total).

The following key themes were mentioned in both focus groups: patients and MHNs in practice regarding oral health; preferred oral health tools, particularly for MHNs working with patients with psychotic disorders; and intended outcomes of tools. It is important to note that themes were interlinked (e.g., context was related to tools; tools were related to context and outcomes). 

Key theme 1: Patients and MHNs in practice regarding oral health

Participants in both focus groups indicated that patients frequently experience dental issues. The expert by experience stated that many patients with psychosis do not pay much attention to oral health at all. Patients do not contemplate oral care (or self-care) and are genuinely focused on other matters. 

The expert by experience indicated that when patients have oral problems (e.g., bad breath), nobody comments on them. 

“A patient is not aware of it himself; patients live in their own reality and if patients are ill, they simply are not always aware of it. MHNs need to bring this up for discussion”. (Participant 9, MHN) 

MHNs indicated that many patients with psychosis use medication. The influence of medication on oral health is known by some MHNs, but this is not always communicated to patients, suggesting that this should be considered when developing a tool. Participants stated that when patients need to visit the dentist, it incurs a significant cost, and they often do not have dental insurance. Consequently, patients are left with expensive bills, and they often do not have the necessary funds available.

Participants asserted that MHNs do receive information about toothbrushing during their education, but it ends there. The dental hygienist pointed out the importance of background information. It is crucial for MHNs to have background knowledge.

Furthermore, MHNs reported not always being aware of what to do in case of a dry mouth or when patients are unwilling or unable to brush their teeth.

MHNs indicated that they are caught up in daily routines and pay little attention to patients’ dental health. Some MHNs did encounter issues but found it challenging to discuss them with patients.

“For example, I also have a patient, and when I speak with her, I can see that she has a lot of dirt and plaque on her teeth. I discuss it with her every time, but she keeps saying, ‘No, it’s going well, and I will go to the dentist,’ and I find that challenging. I try to bring up the topic of oral care again. However, I notice that she gets annoyed when I discuss it repeatedly”. (Participant 6, MHN) 

Key theme 2: Preferred oral health tools, focusing on MHNs working with patients with psychotic disorders

Participants discussed the tools that should be developed for MHNs to be sensitive in their support of patients maintaining and improving their oral health. However, they also mentioned that tools targeted at MHNs are also indirectly linked to tools aimed at patients with psychotic disorders and vice versa. 

“If MHNs do not have a clear understanding of the background of tools targeting patients with a psychotic disorder, MHNs cannot address, advise, or monitor them effectively”. (Participant 4, MHN/lecturer) 

In both focus groups, a wide range of tools to support MHNs in maintaining oral health in patients with psychotic disorders was suggested. 

MHNs stated that it is their responsibility to remind patients about daily routines, as mere verbal reminders may not prompt action from patients. Participants suggested developing a reminder in the form of a sticker that patients can place on a mirror (tool 3). Indirectly, MHNs will also see this and then address the issue. Participants stated that it is important for patients with psychotic disorders to have at least a toothbrush, toothpaste, and interdental brushes (tool 2). They suggested providing these in a nice toiletry bag (focus group 2) or a small box (focus group 1). This applies to both inpatients and outpatients.

“If we provide this to every patient when they start receiving care, it also prevents us from discovering too late that the materials are not available. This is crucial because patients with a psychotic disorder often do not think to bring these items. If we routinely include this during the anamnesis, it will not be forgotten by MHNs”. (Participant 2, MHN) 

The dental hygienist emphasised the importance of patients having mouthwash (tool 2). This is especially significant as the 2-min toothbrushing routine can sometimes be a significant task. If it proves to be genuinely challenging at any time, they can use mouthwash. However, it is important to note that mouthwash is not a substitute for toothbrushing; bad breath is caused by the presence of bacteria in the mouth that produce odours. 

“Mouthwashes can aid in combating these bacteria and temporarily freshening the breath. Additionally, it is important for the toothpaste to contain fluoride, as fluoride is known to protect the teeth against acids and sugars”.(Participant 5, oral health hygienist)

Participants discussed the importance of narrative stories (e.g., on problems with oral health or the benefits of good oral health), as provided by experts by experience (tool 1). Members of both focus groups suggested that these narrative stories could be recorded in a video or be part of a tutorial in which an MHN engages in a conversation with an experienced expert. The discussion should focus on the details of what happened and what the expert did to improve a patient’s oral health. 

“Yes, that’s a great idea, and let’s especially emphasize the benefits of good oral health to keep it positive”. (Participant 1, experienced expert/communication specialist)

The participants in both focus groups emphasised that the videos are also beneficial for MHNs as these direct experiences can lead to increased sensitisation and knowledge. Members of both focus groups suggested organising clinical lectures (tool 8) for students in nursing education programmes in which experts by experience could share their narratives. Allowing these individuals to elucidate the significance of proper oral care and its impact on patients’ lives could raise the sensitisation of prospective nurses regarding the importance of addressing this matter.

Participants in both focus groups stated that posters (tool 12) may increase awareness among MHNs as well as patients. Posters could be hung up in the Waiting Room/Office Living Room/Hallway/Drop-in Centre for Homeless Individuals. Here, a limitation was seen: they often have to be replaced to attract attention. 

Participants in both focus groups proposed developing daily schedules with patients (tool 9); MHNs and patients with psychotic disorders could create a schedule together. In addition to activities such as bathing, dressing, and eating, oral care could also be scheduled within this routine. This daily schedule would ensure that both patients and MHNs have a reminder, making them both aware of its importance. The dental hygienist emphasised the importance of reintegrating dental care into patients’ daily routines.

Members of both focus groups suggested that it would be beneficial to develop a brochure to provide to patients with psychotic disorders upon admission (tool 4). When asked about the content of the brochure, participants indicated that it should contain information about the importance of oral care, risk factors, oral complaints, physical symptoms, psychological symptoms, shame, fear, medication side effects, and dental consequences, as well as nutrition. Additionally, members of focus group 2 suggested that the brochure should include a detachable awareness screener (tool 5) for MHNs to revisit later. This awareness screener should include questions about patients’ satisfaction (or dissatisfaction) with their oral health, what changes they would like to make, and what they would need in order to make those changes. 

“In the hospital, you would also have an awareness screener to verify if the brochure has been distributed and discussed, enabling a later follow-up. These are the focal points”. (Participant 8, MHN) 

Furthermore, participants in focus group 2 indicated that an engaging, positive, light-hearted, and humourous video (tool 6) could potentially motivate MHNs to place greater emphasis on oral care.

Participants in focus group 1 deemed it essential for nurses to have an instrument that provides insight into the current oral care of patients with psychotic disorders (e.g., OHAT) (tool 14). Additionally, participants in focus group 1 discussed OHIP (tool 13) as a brief assessment tool where oral health-related quality of life is assessed. This might give MHNs insight into patient experiences and impacts. 

Participants indicated that it would be important to have all the information gathered in one place on a website (tool 15) (focus group 1). It would be advantageous to have a QR code on a card, enabling scanning for quick access to the information. Participants stated that this might also contribute to the sustainability of integrating tools to help MHNs focus on oral health care. The website could also incorporate supplementary videos, such as instructional videos pertaining to oral care (tool 7). Participants (focus group 1) also noted that there already exists online information on tooth brushing, such as a reference card on not overlooking the mouth (tool 10) (in Dutch: Zakkaartje de mond niet vergeten; in English: Pocket card: Do not forget oral care) and a reference card on brushing techniques (tool 11) (in Dutch: Zakkaartje poetsen doe je zo; in English: Pocket guide: How to clean properly).

Key theme 3: Intended outcomes of tools

All participants in both focus groups expected behavioural changes in MHNs regarding their focus on oral health care and oral disease prevention. The intended outcomes of the tools were discussed in both focus groups. Participants noted that the tools outlined in the paragraph above would enhance the sensitisation and knowledge of MHNs due to the availability of information and materials and would enable MHNs to engage in conversations with patients with psychotic disorders. Participants stated that these should serve as the primary outcome measures. Furthermore, participants said that it is essential that the tools be sustainably implemented and not intended for one-time use. 

### 3.2. MCDM Results: Step 2

Step 2 took place in October 2023. The participants (*n* = 9) assessed all 15 tools derived from the pressure cooker focus group session (Table 1). Table 2 shows the unweighted total mean scores of tool prototypes in the MCDM (ranked high–low). 

The results in Table 2 indicate that participants preferred that multiple tools be developed within a toolkit for oral health. The results demonstrate that a brochure for MHNs and patients (prototype 4) achieved the highest unweighted mean score (mean 4.0). Although the awareness screener received slightly lower scores (mean 3.7), the research team elected to incorporate an awareness screener into the brochure as this could provide a tangible way to facilitate dialogue (Table 2). 

### 3.3. Development of the Brochure with Awareness Screener and the Outcomes of the Feasibility Test: Step 3

#### 3.3.1. Development of the Brochure with Awareness Screener

As part of a larger toolkit, and based on the highest mean scores, the research team decided to start developing a brochure for MHNs, and the research team elected to incorporate an awareness screener into the brochure as this could provide a tangible way to facilitate dialogue. The brochure with an awareness screener was developed as a low-fidelity prototype based on the latest scientific insights from the literature (by SK, NB, and SB). The brochure was developed after the MCDC.

The content of the brochure covered the following: 1. Introduction: The Importance of Good Oral Health; 2. Psychological and Social Complaints|Stigma; 3. Physical Complaints; 4. Risk Factors; 5. Nutrition; 6. Medications, Side Effects, and their Impact on Dental Health; 7. Problems in the Mouth; 8. Advice: What Advice Can You Give Your Patients?; 9. Awareness Screener (Appendix D and Appendix E). 

The perceived advantage of a brochure is that it can provide a foundational knowledge base which creates sensitisation, enabling MHNs to engage in conversations with patients with psychotic disorders. Furthermore, it is beneficial for MHNs to go through and complete the awareness screener together with their patients. By incorporating a visual analogue scale (VAS) [34] into the awareness screener, care actively involves patients, potentially enhancing the integration of oral health care into routine nursing practices. 

#### 3.3.2. Outcomes Feasibility Test

Step 3a took place in November 2023. For the semi-structured interviews (*n* = 19), 12 professionals and seven experts by experience were included. The total group consisted of 12 women and five men; their ages ranged between 27 and 58 years. Participant characteristics (gender, age, profession, educational level, years of working experience, working with in- or outpatients) are given in Appendix C. Each session had a duration of 30–90 min.

The results were analysed in terms of different aspects of feasibility: acceptability, demand, implementation, practicality, integration, and efficacy [30]. 

Acceptability

All participants expressed high satisfaction with the brochure (Appendix D and Appendix E). The feedback characterised it as clear, engaging, pertinent, aesthetically pleasing, well organised, and informative, particularly regarding aspects previously unknown to nursing professionals. Participants said it was commendable that this now exists and receives attention.

All participants thoroughly scrutinised the brochure along with the awareness screener. Most of them also reviewed the brochure and screener with their colleagues, and two participants reviewed them with a patient. The discussions covered their findings and, based on their input, the content of the brochure was revised accordingly (Appendix D and Appendix E). The brochure included the following sections: (1) Introduction: The Importance of Good Oral Health; (2) Psychological and Social Complaints|Stigma; (3) Physical Complaints; (4) Risk Factors; (5) Nutrition; (6) Medications, Side Effects, and their Impact on Dental Health; (7) Problems in the Mouth; (8) Advice: What Advice Can You Give Your Patients?; and (9) Awareness Screener. 

Participants held various views regarding the language that should be used in a brochure. Two participants thought that the brochure was written in an overly complex manner and thus might be inaccessible to MHNs. They indicated that it primarily concerns others and not themselves. Two participants thought that the language was too simplistic. Other participants found the language to be well tailored to their needs.

All participants said the brochure was highly effective at fostering sensitisation. In particular, the visual analogue scale (VAS), part of the awareness screener, was well suited for this purpose. The VAS is a psychometric measurement tool comprising a linear scale with opposing statements at each end. By asking patients about their satisfaction with their oral care and how they perceive their dental hygiene practices, the scale offered opportunities to initiate dialogue.

Participants stated that the brochure could prompt reflection and moments of realisation among nursing professionals.

“Content-wise, the brochure is exceptionally strong for MHNs. It is a comprehensive brochure covering diverse topics, including general oral care. For myself, it is also beneficial to read. This prompts consideration of personal application… This brochure is useful for everyone to look through. It particularly emphasizes the risks, underscoring the importance of proactive engagement with these issues”. (Participant 9, MHN and somatic screening practitioner)

Participants deemed the brochure suitable for implementation and practical use. The brochure aligns with the norms and values of organisations, notably in terms of recovery-supportive care. It also resonates with the autonomy-enhancing mission of these organisations, where motivational interviewing also plays a significant role.

Demand

The participants acknowledged that, currently, there is no actual demand for this brochure as oral health care is not a consideration for many MHNs, nor for many physicians. They perceived a consistent pattern of the profession reacting slowly to emerging issues. Therefore, participants emphasised the importance of effectively implementing the brochure.

“There is no explicit demand for this; we conduct our work without adequately focusing on oral care. However, this neglect is intrinsically related to the demographic with psychotic disorders. Although the demand might not be apparent, if we as nurses provide the service and remain attentive, then such a brochure, I surmise, would be beneficial in enhancing our alertness to these needs”. (Participant 3, MHN and somatic screening practitioner)

Some participants indicated that the lack of demand is not at all unusual. If no one considers it, then no demand arises. Almost all participants suggested that the brochure, now that it is available and has been modified in terms of content, will attract considerable interest. Furthermore, all participants expressed their intention to actively utilise it in their professional practice. 

Implementation (expected issues)

All participants said that the brochure was suitable for implementation and, consequently, for practical application. They asserted that effective implementation will be critical for its success. However, some participants indicated that it might be challenging to implement in a high intensive care (HIC) environment, where patients often have short stays and return home as soon as their most severe symptoms subside. When patients transition to a different department following a crisis, the brochure could be helpful as they commence their recovery process. Participants had varying opinions regarding the department where the brochure could be implemented. Some participants thought it would be suitable for HIC units, others only for clinical treatment wards. Conversely, a few believed it is not appropriate for long-term care units and should be limited to the HIC setting. However, all participants agreed that the brochure would certainly add value in the outpatient setting.

In discussions with participants, the tools needed for implementation were explored. Participants suggested that the topic of oral care and the brochure should be included in routine somatic screenings, which occur monthly or sometimes annually, to ensure follow-up. Some participants proposed integrating it into patients’ daily schedules, akin to medication routines. They also mentioned its compatibility with the WHO’s International Classification of Functioning, Disability and Health (ICF) framework, though this model has not yet been implemented [35]. Furthermore, all participants agreed that merely placing the brochure in a department is insufficient. They stressed the importance of providing a simultaneous clinical lesson so that the brochure can serve as a reference. Additionally, an eye-catching factsheet could be created to spark curiosity among MHNs. 

Participants indicated that there are circumstances for patients that make it more challenging for them to address their oral care. These include the following: (1) Some patients, due to traumatic experiences or re-experiencing a trauma, may find it intolerable to have a toothbrush in their mouth. (2) Patients with psychotic disorders often use medication that leads to frequent thirst and hunger. They tend to compensate for this by consuming excess sweet beverages and making poor dietary choices. MHNs do not always have control over this, especially in a home setting. Therefore, employing motivational interviewing as a technique with patients is crucial. (3) Patients who are not insured for dental care must bear the costs of treatment themselves. They are often financially incapable of doing so. This also applies to simpler materials such as mouthwash. (4) For MHNs, it is essential to ascertain whether patients possess basic oral hygiene items such as a toothbrush and toothpaste. Participants reported that it is not uncommon for patients to lack these essentials due to their illness.

Almost all experts by experience perceived a distinct role for themselves in initiating discussions about oral care with patients. Experts by experience could also serve as intermediaries between patients and MHNs. In the context of recovery-oriented care, once a patient’s initial crisis has passed, it would be feasible for experts by experience to broach the topic. 

Practicality 

Most participants indicated that there are likely to be no negative effects. However, two participants acknowledged that some MHNs might express reluctance; one remarked, “Oh, now there’s something else added to our plate, as if we don’t have enough to do already” (Participant 1, social psychiatric nurse). 

All participants believed that the brochure was not complex in content and could effectively inform patients. It is expected that MHNs will be able to translate the information from the brochure to meet the specific needs of individual patients.

“Precisely because the brochure effectively emphasizes the importance of oral care, it leads to an increased sensitization among nursing professionals about their role in oral health, making it difficult for them to overlook this aspect any longer”. (Participant 8, mental health nurse and somatic screening practitioner) 

The costs were discussed with the participants. No participants (MHNs and experts by experience) expected the implementation of this brochure to incur a high cost. One participant said the following: 

“Health insurance providers should ideally include preventative oral care in their basic coverage plans. It is peculiar that while lifestyle programs for overweight individuals are offered at no cost, dental care does not receive the same treatment. This is a significant issue, as proper oral care is equally vital for overall health”. (Participant 5, advanced nurse practitioner)

Among the participants, three had a Bachelor of Business Administration in business economics (participants 6, 12, and 17). As the costs were discussed, these participants asserted that oral care should be an integral part of the daily responsibilities of mental health nurses (MHNs) and that they should actively engage in it. When it comes to training personnel, such as with a clinical lesson, this does involve some staff hours. 

“Another important consideration involves key stakeholders, e.g., health insurers. Given the significant costs associated with oral care, particularly for uninsured individuals, it may be prudent to seek contributions from them. Addressing oral care is relevant not only from a health perspective but also from a business standpoint. Additionally, poor dental health can lead to embarrassment and social withdrawal, hindering participation in society and employment opportunities. This is where local governments can also play a role, potentially by subsidizing costs”. (Participant 12, master’s student in advanced nursing) 

Integration

The sustainability of the brochure was a recurrent theme in all discussions. To ensure enduring impact, it is crucial to have an accompanying clinical lesson. Furthermore, participants deemed it important to have a staff member (or innovator) within a department or organisation acting as a driving force to conduct the clinical lessons and ensure that the topic is discussed during team meetings. During the discussions, the concept emerged of developing a “train the trainer” programme for oral health in mental health. Ownership of a topic is important for motivation and stimulation and can be a point of contact. 

“Nurses who take ownership of a specific topic tend to work with greater dedication and enthusiasm”. (Participant 12, master’s student in advanced nursing)

Efficacy

All participants agreed that the brochure, accompanied by an awareness screener, contributes to MHNs’ sensitisation and knowledge regarding oral health. Additionally, it can help provide education to patients, thereby facilitating discussions about oral care. Whether this will be achieved depends on the implementation, which is the foundation for everything else. 

More than half of the participants stated that it is crucial to discuss the brochure with patients so that MHNs can tailor the content of the education to meet the needs of the individual patient. This implies that, with adequate attention paid to implementation, a certain degree of efficacy might also be anticipated. 

All participants acknowledged a renewed sensitisation of the importance of oral care and affirmed that they will certainly incorporate and apply this knowledge from now on when the situation requires it.

## 4. Discussion

This study is part of a larger research project aimed at developing a toolkit comprising various tools for MHNs and oral care. To the best of our knowledge, this is the first study with a human-centred scientific approach involving a collaboration between MHNs and experts by experience to develop a tool that serves the dual purpose of (1) raising sensitisation among MHNs and (2) equipping them with knowledge on maintaining oral health in patients with psychotic disorders. 

The results of the discussions with MHNs show the following: (1) Of the 15 suggested tools, a brochure with an awareness screener regarding oral health in patients with psychotic disorders is considered the most feasible that would contribute to the sensitisation and knowledge of MHNs. (2) Significant additions to the brochure would include a clinical lesson and the provision of materials (such as a toothbrush and toothpaste) to ensure that all patients are able to follow up on the given advice. (3) There are diverging opinions concerning the applicability of the brochure among various populations of patients. (4) There are different preferences regarding the level of the language in which the brochure for MHNs should be written. (5) The costs of oral health prevention (e.g., materials and dental hygienist visits) might be significant, and usually fall to patients who, in many cases, cannot afford it. 

To elaborate on the first finding, in the pressure cooker session, MHNs and experts by experience identified 15 tools as essential to have in a toolkit for initiating oral care activities. Based on the MCDM, a brochure with an awareness screener regarding oral health in patients with psychotic disorders was the most appropriate tool to start with. This is in line with our earlier research, in which we found that MHNs need more education to effectively include physical health promotion (such as oral health) in their activities [21]. Barik et al. [36] found that traditional health promotion media such as brochures are useful, especially for adults. As part of a larger toolkit, the brochure with an awareness screener was found to be feasible and could contribute to the sensitisation and knowledge of oral health among MHNs. The results indicate that MHNs will engage with the brochure’s content and intend to take action to change their behaviour around patients’ oral health. This suggests that the brochure with its awareness screener could help MHNs progress from pre-contemplation (stage 1) and contemplation (stage 2) to preparation (stage 3) and action (stage 4), according to Prochaska and DiClemente’s behavioural change model [22]. 

Furthermore, implementation within organisations is indispensable as the next step of the Prochaska and DiClemente behavioural change model (step 5) [22] so that MHNs will adhere to their change of behaviour. Here, it is important to note that insufficiently implemented interventions are less effective, causing MHNs to regress to their previous behaviours. The Fogg behavioural model [37] states that people can only achieve behavioural change when three elements are present simultaneously: motivation, ability, and a trigger. This means that when MHNs are motivated to actively change, as shown in this study, they need a trigger, which is something or someone that encourages individuals to act when their motivation is high but their ability is low. These triggers could be innovators or early adopters [38] who prompt action by modelling the behaviour, making it easier for others to initiate. MHNs could be those facilitators when they take professional leadership. In the literature, professional leadership is defined in terms of personal leadership, such as being proactive, being a role model, taking initiative, being self-reflective, showing assertiveness or courage, and focusing on good cooperation [39]. The question of how MHNs can take a professional leadership role and take responsibility for their role is highly relevant in this regard [21]. On the one hand, this starts with sensitisation, knowledge, responsibility for acting, and actual implementation, and the brochure provides a foundation for this purpose. On the other hand, it also requires individuals who demonstrate professional leadership to actively engage with the brochure, specifically concerning the topic of oral health. To improve ability, the suggested clinical lesson should be developed in an easy and accessible way to make it a worthwhile investment in terms of both content and time [37]. The adequate implementation of interventions in organisations is a highly relevant issue but was not within the scope of this study. Regarding the expected effects, definitive statements cannot be made, but the research can be continued to explore this further.

To elaborate on the second finding, notable enhancements to the brochure would include providing a clinical lesson and offering patients a toothbrush and toothpaste. This suggests that additional tools and materials in a toolkit are needed to facilitate the implementation of interventions in a manner that is both actionable and potentially sustainable. Based on the present study, we cannot (yet) say anything about the actual effectiveness of these tools regarding knowledge and awareness about oral health among MHNs. Further research into the design of a more comprehensive toolkit to help MHNs with maintaining and improving oral health in patients with psychotic disorders is indicated.

The third finding was that there are diverging opinions concerning the applicability of the brochure with the awareness screener when considering various populations. The content of the brochure was developed for MHNs who care for patients with psychotic disorders. A recent review shows that PTSD is a common problem among people with psychosis, with a prevalence of 14–47% [40]. The prevalence of trauma focused on the face or mouth is not known. When trauma is focused on the face or mouth, it might be important to develop alternative strategies in collaboration with oral care specialists. This also applies to patients with, e.g., eating disorders who frequently regurgitate and have other needs regarding oral health [41]. The prevalence of anorexia nervosa in patients with psychotic disorders has been approximated to be between 1 and 4% [42]. However, even though this pertains to relatively small groups, the expectation is that the brochure will be useful in most cases.

The fourth finding was that there are diverging views concerning the level of language proficiency at which a brochure ought to be composed. A general recommendation is to write a brochure at the CEFR B1 level, which means using language that is understandable to virtually everyone [43]. This presents a complexity, however. The brochure is tailored to a target group: MHNs. In the Netherlands, MHNs operate at three distinct levels: secondary vocational education in nursing, bachelor’s level in nursing, and master’s level in nursing. Given these three distinct levels, for the purposes of this study and in the absence of explicit guidelines, it was decided to align the text with the demographic for whom it should be most easily comprehensible (i.e., CEFR B1).

Despite the fifth finding of our study, that oral health care comes with financial costs that patients may not be able to meet, oral health management (e.g., dental visits, dental hygienist visits, materials to maintain oral health) is nevertheless important to prevent problems in later stages of illness as well as physical problems (such as problems in diabetes type 2). During admission, patients should be more adequately evaluated and supplied with essential items (such as toothbrushes, toothpaste, and mouthwash). Currently, these costs fall to patients, unless they have dental insurance (in the Netherlands). Previous research has demonstrated that many patients are unable to afford additional dental insurance [8,23]. This seems to be a bigger social problem that needs more attention. Policy- and decision-makers should consider providing free dental care for people with psychotic disorders, given the importance of oral health to overall health. The government, municipalities, and mental health organisations—together with health insurance agencies—should work on adjusting services regarding insurance plans and alleviating the financial problems of this vulnerable group of young patients. 

### Strengths and Limitations of This Study

Within this study, we applied various populations and methods for triangulation, which enhanced the validity of the findings by combining diverse methods. This approach is crucial for mitigating the inherent biases that can arise when relying solely on a single method, as is often seen in more conventional qualitative research. The broad range of information sources utilised in this study, coupled with the application of the MCDM methodology, provided a robust means of scrutinising tools and enhanced the ecological validity.

The methods used in this study and the results contribute to the pragmatic validity of the brochure with an awareness screener. Pragmatic validity refers to the extent to which it can be expected that the respective action will yield the intended outcome, rather than the actual effectiveness of actions taken [44]. Thus, the outcomes of this research indicate the need for further research.

During the decision-making process in this study, the MCDM approach was adopted. Within the MCDM framework, weights are assigned to various criteria upon which scoring is based [29]. Criteria such as sensitisation and knowledge could potentially be counted twice. However, this was deemed to be not conducive to the decision-making process in this research. Additionally, it was presumed that this approach did not have any influence on the outcomes.

This research was focused on developing and refining a tool. Although numerous professionals and specialists participated in this study, financial experts were not included; but, had they been, they might have provided valuable advice regarding the cost implications. In our feasibility test, costs were discussed; however, the depth of this discussion was limited due to a lack of expertise in this area. Consequently, it is not clear whether this might have biased the results, which means that cost could be considered in future steps building on this research.

All participants expressed enthusiasm for the topic of oral health and promptly responded to the invitation to participate. However, until now, their engagement with the subject has been limited. It is therefore possible that we did not include participants who are not interested in engaging with the topic.

This study was conducted among a sample of MHNs and experts by experience distributed throughout the northern and eastern parts of the Netherlands. While the number of semi-structured interviews in the third phase was modest (*n* = 19), we were confident that we reached data saturation as no new information was retrieved in the last four interviews.

## 5. Conclusions

This study shows the importance of starting with increasing sensitisation among MHNs and providing them with the knowledge necessary to recognise oral health problems in patients with psychotic disorders. MHNs need a toolkit with a combination of different interventions. To start, a brochure with an awareness screener was developed. The brochure is an accessible tool that was developed in collaboration between MHNs and experts by experience and it is immediately applicable in practical settings. The content of the brochure is tailor-made and demonstrates what MHNs require, and it matches the most relevant stages of readiness for behavioural change in MHNs. Future steps to build upon this research, with the aim of further refining and optimising the process by which oral health tools are to be used by MHNs, should refine the eligibility criteria for psychiatric populations and the language level of the target group.

The human-centred design (HCD) process, involving an intensive collaboration between mental health professionals and experts by experience, yielded meaningful results. This study shows that actively involving MHNs and experts by experience in the development process can initiate change. Many MHNs and experts by experience were motivated to test the brochure, which also demonstrates a significant level of engagement with the issue of oral health and that MHNs are willing to act but lack the necessary tools. 

The overall procedure of the research methods used in the development process can be used to develop a variety of tools and demonstrates that not all tools need to be overly complex. This approach provides a foundation upon which to begin implementation and allows MHNs and experts by experience to provide input and feedback to promote engagement, responsibility, and professional leadership, which could enhance the sustainability of the tool. 

## Figures and Tables

**Figure 1 ijerph-21-00492-f001:**
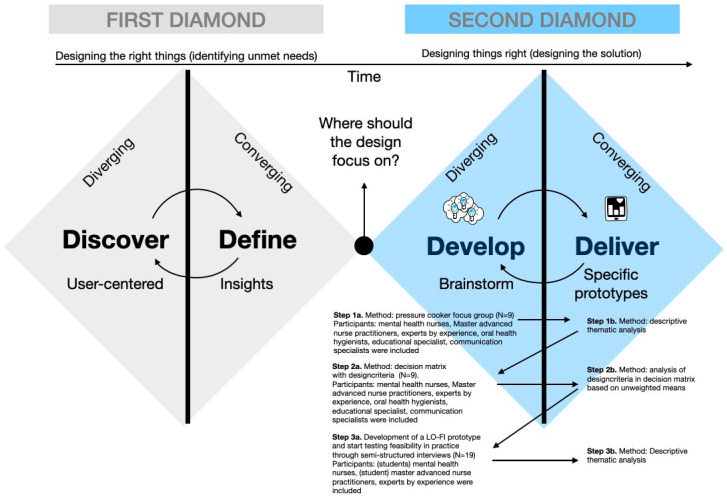
The iterative human-centred design stages and data collection activities used in the second diamond of the double diamond model from the Design Council [24], adapted with permission from van ‘t Veer et al. [26].

**Figure 2 ijerph-21-00492-f002:**
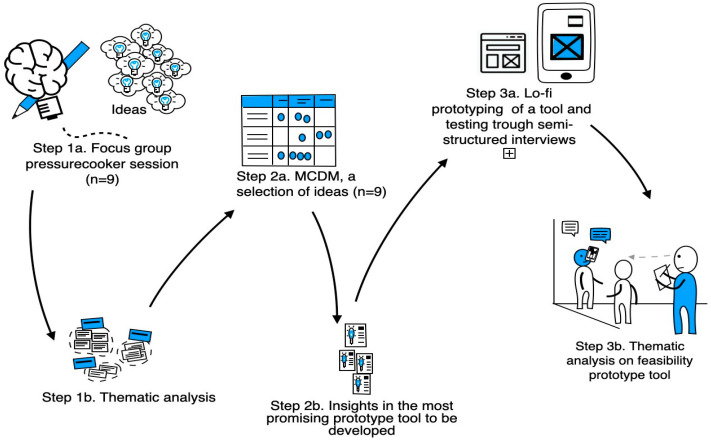
Visualisation of the steps of data collection and data analysis employed in this study.

**Table 1 ijerph-21-00492-t001:** Characteristics of professionals and experts included in steps 1 and 2 (*n* = 10).

Part	Profession	Male/ Female	Age in Years	Education	Years of Working Experience	Focus Group
1	Communication specialist/content maker/expert by experience ^1^	Male	31	Central Institute for Training Sports Leaders	5–9	1
2	Mental health nurse (flexible assertive community treatment)	Male	27	Bachelor of Nursing	0–4	1
3	Education and research consultant	Female	48	Bachelor of Nursing, Master of Educational Sciences	0–4	1
4	Lecturer in nursing research and mental health nurse	Female	53	Ph.D. student, Bachelor of Nursing, Master of Social Work, Master of Health Care	10–14	1
5	Oral health hygienist	Female	25	Bachelor of Oral Health Hygiene, Master of Health Innovation	0–4	1
6 *	Master’s-level advanced nurse practitioner	Female	47	Ph.D., Master of Advanced Nursing Practice	>24	2
7	Team leader of Academy of Health and Out of the Box Thinking	Female	52	Bachelor of Movement Sciences	5–9	2
8 *	Mental health nurse (outpatients)	Female	50	Bachelor of Nursing	>24	2
9	Expert by experience	Female	32	Student, Bachelor of Social Work	0–4	2
10 *	Expert by experience ^2^	Male	48	Associate degree for Expert by Experience	5–9	N/A

^1^ This expert by experience (5) did participate in the focus groups but did not participate in the decision matrix. ^2^ This expert by experience (10) did not participate in the focus groups but did participate in the decision matrix. * Participants were also engaged in earlier studies [21].

**Table 2 ijerph-21-00492-t002:** Total of unweighted mean scores of prototypes of tools in the MCDM ranked based on highest mean (*n* = 9).

Tool	Prototype	Mean	Focus Group
4	Brochure for MHNs and patients: Importance of Oral Care Risk Factors Oral Complaints Physical/Psychological Symptoms Shame/Anxiety Medication Side Effects Dental Consequences Nutrition	4.0	1 and 2
8	Clinical Education Session involving the presentation of experiential narratives to students by experiential experts (successful stories).	3.9	1 and 2
15	A website where you can access all information easily and conveniently through a QR code.	3.9	1
5	Awareness screener as an insert in the brochure (prototype 4) for future reference: Assessing your satisfaction (or dissatisfaction) with your oral health, identifying desired changes, and determining the necessary steps.	3.7	2
11	Pocket Card: This Is How You Brush Your Teeth. A compact card illustrating oral care practices.	3.6	1
13	Assessment of the Impact of Oral Health Care on Quality of Life (OHIP 14): A brief, validated questionnaire comprising 14 questions regarding perceived oral health over a one-month period.	3.6	1
14	Oral health screening:(validated for non-dental professionals in Dutch).	3.5	1
7	Instructional Video for Patients and MHNs on How to Brush and Floss Teeth.	3.5	1 and 2
1	Video: Nursing Professional and Expert by Experience: A Discussion of Personal Narratives Concerning Oral Health Care. This encompasses an exploration of first-hand experiences relating to oral health care. Queries addressed include the nature of these experiences, the methodologies employed for managing them, the status of the involved individuals’ oral health, and the outcomes or benefits that have been realised.	3.4	1 and 2
3	Mirror-Based Dental Care Reminder Sticker.	3.4	1 and 2
9	Daily Schedule for Patients: In this daily schedule, patients and nursing staff collaboratively document activities such as waking up, breakfast, medication, and toothbrushing.	3.4	1 and 2
2	A toiletry bag or box (M/F) containing the following items: Toothbrush Toothpaste (containing fluoride) Dental floss picks Mouthwash Plaque-disclosing tablets (a literature-supported pill that highlights dental plaque)	3.3	1 and 2
12	Poster for Waiting Room/Office Living Room/Hallway/Drop-in Centre for Homeless Individuals.	3.3	1 and 2
10	Pocket Card: Do not Forget Your Oral Health. This card facilitates discussion about current oral care practices (self-care, dental visits, dental condition).	3.2	1
6	Toren C Instructional Video on Toothbrushing that is light-hearted, positive, and very humourous.	2.9	2

## Data Availability

The data that support the findings of this study are available upon request from the corresponding author.

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
