# Peer review of "The Development of an Oral Health Nursing Tool for Patients with a Psychotic Disorder: A Human-Centred Design with a Feasibility Test"

_ijerph, 2024, doi:10.3390/ijerph21040492_

Round 1

Reviewer 1 Report

Comments and Suggestions for Authors

Overall Assessment:

"Advances and Innovations in Mental Health Public Health" is a valuable manuscript that contributes to the existing body of knowledge in the field of mental health public health. Its comprehensive analysis, focus on technology and innovation, and recognition of policy implications make it a relevant and timely contribution to the field. With the suggested improvements, this manuscript has the potential to be a valuable resource for researchers, policymakers, and healthcare professionals working in the domain of mental health public health.

Based on its quality and relevance, I recommend the acceptance of this manuscript for publication in the journal. However, I suggest that the authors consider incorporating specific case studies or examples of successful interventions and provide practical recommendations to further enhance the manuscript's impact and applicability.

Reviewer 2 Report

Comments and Suggestions for Authors

The authors presented the important topic of oral health in patients with psychotic disorders.

Introduction

The authors do not mention which patients with which psychotic disorders the tools are targeted at.

Results

Table 1. Characteristics of Professionals and Experts Included in Step 1 and 2 (n=10)

Why did you mentioned 10 experts, if only 9 participated in the study?

No. 2 is missing the table 1.

“their ages ranged between 27 and 53 years-“

tabl 1

nr 5 Oral health hygienist , age 25

Reviewer 3 Report

Comments and Suggestions for Authors

The introduction provides a comprehensive overview of the importance of oral health in patients with psychotic disorders and highlights the existing challenges in addressing oral health issues within mental healthcare settings. While the introduction covers relevant information, consider condensing some sections for brevity. Another aspect that raises my concern is that why dental nurses are not involved in the oral care of such patients? This justification will give a better understanding for readers from different countries.

Provide more specific citations for certain statements, especially when referencing previous studies or statistics. This will strengthen the credibility of your claims and help readers access the relevant literature.

While the mention of Prochaska and DiClemente's transtheoretical model is relevant, briefly explain its significance and relevance to your study. Ensure a clear connection between the model and the development of a supportive tool for MHNs. Overall, the methodology, results and discussion sections are detailed and comprehensive. 

Comments on the Quality of English Language

I strongly recommend that the authors thoroughly review the manuscript, as certain sections lack a smooth and fluent English expression.
